# Study of the Embryonic Toxicity of TiO_2_ and ZrO_2_ Nanoparticles

**DOI:** 10.3390/mi14020363

**Published:** 2023-01-31

**Authors:** Elena Nikolaevna Lyashenko, Leyla Djavadovna Uzbekova, Valeri Vladimirovna Polovinkina, Anastasia Konstantinovna Dorofeeva, Said-Umar Sithalil-ugli Ibragimov, Arslan Ayavovich Tatamov, Albina Gamdullaevna Avkaeva, Anastasia Alekseevna Mikhailova, Inga Shamilevna Tuaeva, Ruslan Kazbekovich Esiev, Sergei Dmitrievich Mezentsev, Marina Alexandrovna Gubanova, Natalya Grigorevna Bondarenko, Alina Yurievna Maslova

**Affiliations:** 1Department of Obstetrics and Gynecology, Faculty of Pediatrics, S.I. Georgievsky Medical Academy, V.I. Vernadsky Crimean Federal University, 295007 Simferopol, Russia; 2Faculty of Pediatrics, Rostov State Medical University, 344009 Rostov-on-Don, Russia; 3Faculty of Medicine, Dagestan State Medical University, 367029 Makhachkala, Russia; 4Faculty of Medicine, Stavropol State Medical University, 355017 Stavropol, Russia; 5Department of Hygiene, Faculty of Medicine and Prevention, North Ossetian State Medical Academy, 362019 Vladikavkaz, Russia; 6Department of Dentistry, North Ossetian State Medical Academy, 362019 Vladikavkaz, Russia; 7Department of the National Research, Moscow State University of Civil Engineering, 125009 Moscow, Russia; 8Armavir State Pedagogical University, 352901 Armavir, Russia; 9Department of Philosophy of History of Law, Pyatigorsk Branch of North Caucasus Federal University, 357502 Pyatigorsk, Russia; 10SocMedica, 121205 Moscow, Russia

**Keywords:** titanium, zirconium, safety, nanoparticles, pregnancy, photon correlation spectroscopy, embryotoxicity

## Abstract

Currently, the widespread use of TiO_2_ and ZrO_2_ nanoparticles (NPs) in various industries poses a risk in terms of their potential toxicity. A number of experimental studies provide evidence of the toxic effect of TiO_2_ and ZrO_2_ NPs on biological objects. In order to supplement the level of knowledge and assess the risks of toxicity and danger of TiO_2_ and ZrO_2_ NPs, we decided to conduct a comprehensive experiment to study the embryonic toxicity of TiO_2_ and ZrO_2_ NPs in pregnant rats. For the experiment, mongrel white rats during pregnancy received aqueous dispersions of powders of TiO_2_ and ZrO_2_ NPs at a dose of 100 mg/kg/day. To characterize the effect of TiO_2_ and ZrO_2_ NPs on females and the postnatal ontogenesis of offspring, a complex of physiological and biochemical research methods was used. The results of the experiment showed that TiO_2_ NPs as ZrO_2_ NPs (100 mg/kg per os) cause few shifts of similar orientation in the maternal body. Neither TiO_2_ NPs nor ZrO_2_ NPs have an embryonic and teratogenic effect on the offspring in utero, but both modify its postnatal development.

## 1. Introduction

The rapid development of nanotechnology has put the scientific community in front of the need to control the safety of products obtained with their use, the production volumes of which are progressively increasing. Timely assessment of the toxicity and danger of nanomaterials can become a reliable barrier to the unhindered penetration into the life of harmful to human health and environmentally hazardous products and/or contribute to the development of a system of measures to protect against their adverse effects [1,2,3,4,5]. It should also be taken into account that any information shedding light on the peculiarities of the biological action of nanomaterials is of scientific interest per se [6,7,8].

Nanomaterials’ environmental exposure represents a real public health issue that will become highly relevant in the near future when people’s consciousness of this issue increases [9]. In this regard, recently, many works have been directed at studying the influence of various nanoparticles on the viability of biological objects and human health. Thus, Sierra et al. (2016) declared that nanomaterials exposure might lead to epigenetic effects: changes in DNA methylation, histone post-translational modifications, and noncoding RNAs in mammalian cells [10]. Ellis et al. (2021) found that multigenerational exposure to nanosized TiO_2_ induces aging as a stress response mitigated by environmental interactions [11]. Das et al. (2016) demonstrated the toxicological effects of nanoparticles in mammalian germ cells and developing embryos by considering both in vitro and in vivo experimental models [12]. Ma et al. (2022) illustrated that subacute exposure to SiNPs could trigger autophagy dysfunction and pulmonary inflammation in rats [13]. Other researchers discovered that nanoparticles could inhibit epithelial-mesenchymal transition (EMT) [14], change human serum metabolites [15], cause chronic inflammations [16,17], affect macrophage polarization [18], exacerbate atherosclerosis [19], induce pulmonary fibrosis [20,21], and increase cancer risk [22].

Currently, nanocrystalline oxide materials, which have unique properties and a wide range of applications, are attracting increasing attention as objects of research. Such compounds include titanium dioxide (TiO_2_) and zirconium dioxide (ZrO_2_). Being chemically stable and relatively inexpensive, they are widely used for the creation of gas sensors, dyes, protective coatings, dielectric ceramics, thermal insulation materials, hydrophobic materials, photocatalysts, electrochromic displays, fiber optics, solar panels, cosmetics, and medical products, including implants and scaffolds [23,24,25,26,27,28,29,30,31,32]. Thus, TiO_2_ and ZrO_2_ NPs are often used as part of innovative materials and as part of micromachines constantly used by people, which necessitates a reliable, in-depth study of the effect of these nanoparticles on human health.

Due to the widespread use of TiO_2_ and ZrO_2_ NPs, one of the first hygienic standards for nanomaterials was the approximate safe level of exposure for these nanomaterials in the air of the working area [33]. However, the possibility of massive ingress of nanoparticles into wastewater, water resources, and soil during the operation and disposal of products containing TiO_2_ and ZrO_2_ NPs gives no less urgency to the problem of assessing the consequences of its ingestion by oral route [34]. For instance, Liu et al. (2018) declared that TiO_2_, SiO_2,_ and ZrO_2_ NPs synergistically provoke oxidative damage to the algal cells [35]. Farkas et al. (2015) showed that exposure concentrations of 100 μg/L TiO_2_ NP cause significant reductions in bacterial abundance in fresh water [36]. Ye and Shi (2018) detected the cytotoxicity of TiO_2_ and ZrO_2_ NPs in osteoblast-like 3T3-E1 cells and found that reactive oxygen species played a crucial role in the TiO_2_ and ZrO_2_ NP-induced cytotoxicity with concentration-dependent manner [37]. Yang et al. (2019) detected that 100 mg/kg ZrO_2_ has no significant effect on the morpho-functional and biochemical indexes of mice [38]. However, the authors pointed out that at higher doses, ZrO_2_ causes oxidative damage in the liver and serum of mice. In similar studies, the toxic effect of TiO_2_ and ZrO_2_ NPs on laboratory rats and mice was detected at lower concentrations (60–100 mg/kg) [39,40,41,42].

Recent works state that pregnant women and the fetus during the gestational period (GP) are most susceptible to the toxic effect of nanoparticles [43,44,45]. For instance, Teng et al. (2021) showed that maternal exposure to NPs during pregnancy led to adverse gestational parameters, neurotoxicity, reproductive toxicity, immunotoxicity, and respiratory toxicity in rat offspring [46]. Therefore, when considering TiO_2_ and ZrO_2_ NPs, it is especially important to assess their effect on pregnant women and embryo development. The literature review showed a relatively small number of works in this field. Mao et al. (2019) showed that TiO_2_ NPs induce the alteration of gut microbiota during pregnancy and increase the fasting blood glucose of pregnant rats, which might increase the potential risk of gestational diabetes in pregnant women [47]. El Ghareeb et al. (2015) indicated that the intraperitoneal injection of TiO_2_ NPs to pregnant and lactating rat mothers impairs many of the morphological and skeletal formations, as well as vital organs such as the brain, liver, and kidneys [48]. Yamashita et al. (2011) showed that TiO_2_ NPs with a diameter of 35 nm could cause pregnancy complications when injected intravenously into pregnant mice [49]. The authors supposed that detrimental effects are linked to structural and functional abnormalities in the placenta on the maternal side. The same conclusion was made by Wang et al. (2021), who discovered that ZrO_2_ NPs are able to cross multiple biological barriers and were accumulated in the maternal placenta and fetal brains [50]. Similarly, Zhou et al. (2019) suggested that at maternal exposure, TiO_2_ NPs crossed the blood–fetal barrier and blood–brain barrier and deposited in the brain of offspring, which retarded axonal and dendritic outgrowth, including the absence of axonal outgrowth, and decreased dendritic filament length, dendritic branching number, and dendritic spine density [51]. It is important to note that maternal exposure of mice to TiO_2_ NPs may affect the expression of genes related to the development and function of the central nervous system, as was reported by Shimizu et al. (2009) [52]. Lee et al. (2019) studied the effect of TiO_2_ NPs on pregnant rats on gestation days (GD) 6–19 at dosage levels of 0, 100, 300, and 1000 mg/kg/day [53]. The authors did not find acute toxic effects on pregnant rats in the experiment, but titanium contents were increased in the maternal liver, maternal brain, and placenta, which can affect the embryo.

Thus, the current level of knowledge about the effect of TiO_2_ and ZrO_2_ NPs on pregnant women and embryo development has shown that these nanoparticles do not cause a direct negative impact on the gestation but are accumulated in the maternal and fetus organs and can have a significant impact on the further development of the fetus. Nevertheless, the described works were aimed at studying a narrow range of indicators of experimental animals, which do not fully explain the mechanism of the effect of TiO_2_ and ZrO_2_ NPs. Also, as far as we know, no one has conducted a comparative analysis of the toxic effects of TiO_2_ and ZrO_2_ NPs during pregnancy. Therefore, in order to supplement the level of knowledge and assess the risks of toxicity and danger of TiO_2_ and ZrO_2_ NPs, in our opinion, it is necessary to conduct a comprehensive experiment to study the embryonic toxicity of TiO_2_ and ZrO_2_ NPs. The aim of this work was to study the embryotoxic effect of TiO_2_ and ZrO_2_ NPs on pregnant laboratory rats. The novelty of this work is associated with a comprehensive parallel assessment of the effect of TiO_2_ and ZrO_2_ NPs on pregnant rats and fetuses in the same experiment under the same conditions.

## 2. Materials and Methods

### 2.1. Synthesis of TiO_2_ NPs

In the first stage, 190 cm^3^ of distilled water (H_2_O) and 10 cm^3^ of a 25% solution of ammonia NH_4_OH (Lenreactiv, Saints Petersburg, Russia) were measured with a measuring cylinder. The resulting solution was poured into a beaker and mixed on a Topolino magnetic stirrer (IKA, Moscow, Russia) for 2 min.

In the second stage, 10 cm^3^ of a concentrated aqueous solution of titanium tetrachloride TiCl_4_ (Lenreactiv, Saints Petersburg, Russia) was measured with a measuring cylinder and, with intensive stirring, it was poured into an aqueous solution of ammonia, resulting in a white precipitate of TiO_2_⋯nH_2_O. Then a 25% solution of ammonia NH_4_OH (Lenreactiv, Saints Petersburg, Russia) was added drop by drop to a certain pH value. The process was completed at 6 ≤ pH ≤ 8 since it is a gel-formation point. The resulting gel was left in a dark place at room temperature for a day.

In the third stage, the resulting gel was washed 4 times with distilled water and centrifuged using a TSLN-16 centrifuge (Polycom, Moscow, Russia) at 2000 rpm for 6 min. This procedure was carried out to remove impurity ions (Cl^−^, NH4^+^).

In the fourth stage, the washed TiO_2_ gel was dried at a temperature of 125 °C.

### 2.2. Synthesis of ZrO_2_ NPs

In the first stage, 26.7 g of dihydrous zirconium nitrate ZrO(NO_3_)_2_⋯2H_2_O (Lenreactiv, Saints Petersburg, Russia) was weighed on analytical scales by the method of precise weighting. Then the suspension of zirconium nitrate was quantitatively transferred to a beaker, and 50 cm^3^ of distilled water was added. The resulting mixture was stirred on a Topolino magnetic stirrer (IKA, Moscow, Russia) for 30 min and then carefully filtered to remove the insoluble components of zirconium.

In the second stage, a 12% vol. aqueous solution of ammonia NH_4_OH (Lenreactiv, Saints Petersburg, Russia) was added to a filtered aqueous solution of zirconium nitrate at a concentration of 1 mol/L with intensive stirring by a mechanical stirrer drop by drop until pH = 4, at which the formation of a stable gel was observed.

In the third stage, the gel was centrifuged using a TSLN-16 centrifuge (Polycom, Moscow, Russia) at 2000 rpm for 6 min and washed three times with distilled water to remove the impurity ions remaining after the reaction.

In the last stage, the resulting gels were dried and calcined at various temperatures from 125–850 °C in order to remove the remaining water and to decompose zirconium hydroxide to zirconium dioxide.

### 2.3. Research of TiO_2_ and ZrO_2_ NPs

Identification of TiO_2_ and ZrO_2_ NPs was carried out by X-ray phase analysis with PANalytical Empyrean installation (Malvern Panalytical, Malvern, UK). The analysis and preparation of drugs were carried out according to works [25,26]. The identification of diffractograms was carried out according to another work [24]. Measurements of NPs sizes were carried out using a Photocor Complex installation (Antek-97, Moscow, Russia). Computer processing of spectroscopy results was carried out using DynaLS software (Antek-97, Moscow, Russia) [54].

### 2.4. Determination of TiO_2_ and ZrO_2_ NPs Embryotoxicity

The scheme of the experiment with pregnant rats is presented in Figure 1.

In the experiments, the dispersion of the initial samples of TiO_2_ and ZrO_2_ NPs obtained ex tempore by diluting the powder with deionized water was used. Aggregation of NPs was prevented by sonification of the solution using Megeon 76,010 ultrasonic bath (Megeon, Moscow, Russia).

Mongrel white rats with an initial body weight of 200–240 g and a regular estrous cycle were used as a biomodel. Working solutions of TiO_2_ and ZrO_2_ NPs at a dose of 100 mg/kg/day were administered per os to females using a probe at the rate of 1.0 mL per 100 g of body weight practically throughout the entire gestational period from the 1st to the 20th GD. The exposure dose was determined based on data on the multidirectional toxic effect of TiO_2_ or ZrO_2_ NPs reported in [47,48,49,50,51,52,53]. Parallel control animals received deionized water in an equivalent volume daily. The 1st GD was considered the date of detection of spermatozoa in vaginal smears after mating with intact males. Experimental groups (two experimental and one control) were formed according to the cumulative principle, gradually bringing their number to 25 individuals in each for 4 days. For ease of identification, the following designations of experimental groups were introduced: Group 1 (TiO_2_ NPs), Group 2 (ZrO_2_ NPs), and Control. Laboratory animals were placed in individual polypropylene cages KMK-1 (UniZoo, Moscow, Russia) measuring 384 mm × 250 mm × 163 mm. The KMK-1 cages were manufactured in full compliance with all the established requirements and norms [55,56], providing convenient and proper animal maintenance (Figure 2).

When compiling the working algorithm of the experiment, we took into account the uniqueness of the relationships that develop in mammals during pregnancy between the organisms of the mother and fetus and proceeded from the need to apply a systematic approach as an adequate method of their analysis [57]. The last stage was implemented by conducting a simultaneous assessment of the state of the main links of the functional system “mother-fetus” at the end of the 3-week exposure cycle by the tested compounds.

Violations of homeostasis in pregnant females were judged by the results of their one-time (on the 21st GD) comprehensive examination, which was completed by euthanasia and necropsy in half of each group. At the autopsy, a targeted revision of the reproductive organs was performed in females. Fetuses were assessed for general physical development according to the somatometric parameters measured in them. In parallel, indicators characterizing the functional activity of the fetal life support system (weight and diameter of the placenta disc) were recorded. Fetal material seized at autopsy was subjected to special fixation for further microanatomic study.

Taking into account the possibility of manifestation in postnatal ontogenesis of disorders, the foundations of which were laid during the period of development in utero, the second half of the numerical composition of each group of pregnant rats was brought to natural delivery, and the postnatal development of their offspring was monitored.

To identify shifts induced by TiO_2_ and ZrO_2_ NPs, a set of methods were used, the choice of which was determined by the tasks of a particular stage of the experiment and the features of the anatomical and physiological organization of the object of study.

The functional state of the organism in sexually mature individuals was judged by integral tests: body weight, summation-threshold indicator (STI), heart rate (HR), muscle strength, the morphological composition of blood, and behavioral reactions in the automated system of registration and identification of behavioral acts Metris LABORAS [58,59,60]. The effect of TiO_2_ and ZrO_2_ NPs on rat metabolism was evaluated based on the analysis of changes in the main biochemical parameters of blood serum, including total protein content, albumin, glucose, triglycerides, cholesterol, urea, creatinine, lactic and pyruvic acids, the activity of alanine transaminase (ALT), aspartate transaminase (AST), lactate dehydrogenase (LDH) and gamma-glutamyltransferase (GGT) according to the methods described in [61]. The activity of the mechanisms of antiperoxide protection of the body was judged by the content of reduced glutathione in whole blood and the final product of lipid peroxidation (malondialdehyde, MDA) in serum. The studies were performed using the MNCHIP Celercare V5 biochemical analyzer (MNCHIP, Moscow, Russia) and unified spectrophotometric methods [62]. In infant rats and juvenile animals, species-typical indicators of physical development and the rate of maturation of sensory-motor reflexes were monitored [63]. The fetuses were examined for the presence of developmental anomalies by conventional microanatomic methods.

The obtained experimental material was subjected to statistical processing. The indicators are taken into account in an alternative form (conception indices, frequency of anomalies, values of embryonic and postnatal death) and were compared by the Fisher method. Statistical processing of graded features with continuous distribution was carried out using the Student’s criterion using STATISTICA 12 (Statsoft, Tulsa, OK, USA). The differences were considered significant at *p* < 0.05.

## 3. Results

### 3.1. Characterization of TiO_2_ and ZrO_2_ NPs

The structure and phase composition of TiO_2_ and ZrO_2_ NPs samples were determined by X-ray phase analysis. The diffractogram of TiO_2_ and ZrO_2_ NPs samples is shown in Figure 3.

As can be seen from the diffraction pattern (Figure 3), the samples of TiO_2_ and ZrO_2_ NPs are amorphous. There are no characteristic peaks that do not allow for determining the structure and phase composition of the samples. Therefore, we used the method of correlation spectroscopy of dynamic light scattering (photon-correlation spectroscopy) to determine the size of the phases of the prepared samples. Figure 4 shows a histogram of the size distribution of TiO_2_ and ZrO_2_ nanoparticles.

According to the data presented in Figure 4, the average hydrodynamic radius of the obtained TiO_2_ NPs was 87 ± 36 nm, and the size of the ZrO_2_ NPs was 138 ± 24 nm. The results obtained correspond to other works [25,26]. Thus, the nanoscale nature of the synthesized samples was proved by photon correlation spectroscopy, which made it possible to proceed to the next stage of the experiment.

### 3.2. Assessment of the Effect of TiO_2_ and ZrO_2_ NPs on Pregnant Rats and Antenatal Development of Offspring

The experiment showed that the 3-week intake of TiO_2_ and ZrO_2_ NPs into the body during pregnancy causes few, although significant shifts in behavior, peripheral blood composition, and metabolic processes in rats (Table 1), which corresponds to the results of Younes et al. (2015) [39]. In female rats receiving TiO_2_ NPs, only eight significant deviations were revealed: an almost threefold reduction of immobility time, leukocytosis, and cholesterol reduction. A 15% decrease in the blood content of the final product of lipid peroxidation (MDA). TiO_2_ NPs also caused an increase in the duration of grooming, the number of counterclockwise circular rotations, and the concentration of pyruvic acid in the blood. In female rats receiving ZrO_2_ NPs, there were 16 significant changes related to the control group. Most of the detected changes reflected a similar direction of action of TiO_2_ nanoparticles. A certain motor stimulation in female rats of this group was indicated by a 25% increase in the maximum speed of movement. Activation of the antiperoxis protection system was indicated by a pronounced decrease in MDA. ZrO_2_ NPs also caused a reduction of creatinine and leukocyte content and an increase in gamma-glutamyl transferase number (Table 1). Similar effects were reported in relative works [50,53]. Detected changes can be related to TiO_2_ and ZrO_2_ NPs’ effect on the antioxidant enzyme activity of rats [64] and defective mechanisms on metabolism [65]. Special behavioral parameters indicated in pregnant rats from experimental groups are probably associated with the neurotoxicological and neurodegenerative effect of TiO_2_ and ZrO_2_ NPs on the brain and CNS [39,66,67].

The signs of distress detected in females when TiO_2_ and ZrO_2_ NPs entered the body did not, however, affect their ability to reproduce offspring. The conception indices, which amounted to 88% and 96% in the first and the second experimental groups, respectively, did not significantly differ from the control indicator, equal to 92%. A sufficient number of pregnant individuals in all groups allowed them to be very evenly divided into subgroups in accordance with the stages of the study (Table 2).

At the abdominal autopsy of the first half of the female rats, it was found that the tested preparations of TiO_2_ and ZrO_2_ NPs did not have a negative effect on the intrauterine development of the offspring, which corresponds well to the result of Lee et al. (2019) [53,68]. In both experimental groups, the fetus carried out in appearance, and the main indicators of viability (craniocaudal size, body weight) corresponded to their physiological age and did not differ from the offspring of control individuals. The placental fetal life support system, judging by the unchanged metric characteristics, had an adequate degree of functional maturity. The same result was reported in other works [49,69]. There were also no significant differences in the values of antenatal death between experience and control (Table 3).

A lifetime examination of the fetuses did not reveal the presence of gross malformations in them. In the subsequent microanatomic study of the recorded fetal material, teratogenic effects were also not observed. Thus, the enlightenment of soft tissues and subsequent analysis of the fetal bone system (32 in the Control, 33 in Group 1, and 27 in Group 2) did not reveal any deviations in the structure of their skeleton in the experiment, as well as in the rates of ossification of cartilaginous blastemas of various bone formations.

During the study of transversal sections of the body, it was found that the internal organs of the fetuses retained normal topography and their characteristic structure. Rare disorders of embryogenesis without characteristic specificity were recorded in all experimental groups, each of which did not exceed the level of spontaneous anomalies of a similar type in the Control in frequency. Comparative analysis of all experimental groups showed a tendency to increase embryogenesis disorders. However, due to the lack of statistical confirmation of the significance of the revealed differences, all embryogenesis disorders in the experiment were qualified as having no causal connection with the effects of the tested preparations of TiO_2_ and ZrO_2_ nanoparticles (Table 4).

Thus, under the described experimental conditions, both TiO_2_ and ZrO_2_ NPs caused a few pathological shifts in the functional mother-fetus system, concentrated within the maternal organism. The absence of differences between the experiment and the control in the frequency and nature of malformations testified against the presence of teratogenic effects in them. Similar results are declared by Jia et al. (2017) [42] and El Ghareeb et al. (2015) [48].

### 3.3. Study of the Effect of TiO_2_ and ZrO_2_ NPs on Postnatal Development of Offspring

The condition of the offspring obtained from the second half of the female rats was monitored for 2 months after birth. Dynamic monitoring of its development included accounting for mortality, registration of morphological maturation parameters, and the study of the rate of formation of basic sensory-motor reflexes during the lactation period. At the age of 2 months, the animals were subjected to a comprehensive examination. With the exception of body weight, which was determined in all rats, the remaining indicators were recorded selectively. To ensure the representativeness of samples (at least 24 individuals) from each individual female, two rats (female and male) were randomly selected for every two to three tests, which made it possible in this case to achieve almost total inclusion of the entire grown population in the experiment.

The observation showed that the development of offspring in experimental groups occurs with some differences. First of all, this concerns such indicators as postnatal mortality and body weight gain, which are fundamental for judging the nature and severity of the action of xenobiotics [53]. It was found that TiO_2_ NPs have some stimulating effect on the increase in body weight by rats during lactation. This indicator in Group 1 was significantly increased relative to the Control on the 14th and 21st days of life. Postnatal mortality had dynamics similar to those observed in the Control (Table 5).

On the contrary, after exposure to ZrO_2_ NPs, the body weight of the rats on the 14th day significantly decreased, and at the same time, there was a tendency to increase postnatal death. Starting from the 21st day, the rise in mortality above the control level by ~2.4 times has acquired a steadily reliable character (Table 5).

Total testing of infant rats testified that the main signs of progressive morphological maturation, such as detachment of the auricle, eruption of incisors, covering with wool, and opening of the eyes, were formed simultaneously in the experimental groups and control. However, the acquisition of gender-specific reproductive traits occurred in different ways, even within one Group, depending on sex. In males, TiO_2_ and ZrO_2_ NPs accelerated the rate of testicular descent, and by the 25th day of life, this process was completed in 83.7% (TiO_2_ NPs) and 78.7% (ZrO_2_ NPs) of males, despite the fact that the same indicator in the Control was only 57.4%. In contrast, the puberty of females exposed to prenatal exposure to ZrO_2_ NPs was delayed. Thus, on the 55th day of life, the opening of the vagina occurred only in 78.3% of females of Group 2, while in the Control, it happened in 98.4% (Table 6).

The assessment of the timeliness and dynamics of the formation of sensory-motor reflexes of the lactation period in experimental animals indicated the desynchronization of the processes of formation of a number of sensory functions. Thus, in the early neonatal period, the experimental offspring demonstrated a better ability to coordinate movements compared to the control, as indicated by higher indicators obtained in both experimental groups in the “flipping on a plane” test (Table 7).

The more developed sense of smell in the descendants of the Group 2 could be judged by a certain advance of control in the timing of the formation of positive responses in the “homing” test. However, along with such changes, there were also negative shifts. In particular, if the descendants of the second experimental group had a percentage of individuals who took the correct pose in the “negative geotaxis” test, the magnitude was comparable to the control indicators. In Group 1, it was significantly reduced on the fifth day of life.

At 2 months of age, chemically induced modifications of the postnatal development of the experimental offspring persisted. Their manifestations were relatively few but biologically significant. In the first experimental group, four main indicators can be highlighted. The first one characterized changes in the sphere of circular motions and was expressed in an increase in the average diameter of the circle of rotations. The second was leukopenia, and the third and fourth were a decrease in the level of total cholesterol and urea in the blood serum (Table 8). The descendants of Group 2, reaching the same maximum speed as the control animals, could not maintain it for as long, and the time of movement with the maximum speed was reduced in comparison with the Control by ~1.6 times. The pyruvic acid level was also lower in Group 2 than in the Control group. Generally, the data obtained correlates with the data presented in Table 1.

Thus, the results of the observation indicate that during the life span from the period of newborn to the age of reproductive maturity, the development of the offspring of Group 1 and Group 2 occurred with a number of deviations from the physiological etalon presented by the parallel Control group. The revealed modifications give grounds to conclude that under the described experimental conditions (100 mg/kg/day), TiO_2_ and ZrO_2_ NPs have an embryotoxic effect which was not reported as a result obtained but was stated as a potential risk in related works [47,48,50,53].

## 4. Conclusions

In this work, for the first time, a comparative analysis of the effect of TiO_2_ and ZrO_2_ NPs on the vital signs of pregnant rats, embryos, and offspring in postnatal development was carried out. It was found that 3-week contact with TiO_2_ and ZrO_2_ NPs (100 mg/kg per os) leads to very few similar homeostatic shifts in pregnant female rats and does not affect their ability to conceive and gestation offspring. The intrauterine development of offspring proceeded without obvious pathological abnormalities: neither an increase in antenatal death nor an increase in structural defects in the fetuses of the experimental groups was recorded. Nevertheless, follow-up of live-born offspring provided experimental evidence that the real consequences of prenatal exposure to TiO_2_ and ZrO_2_ NPs turned out to be more serious.

During the discussion of the results obtained, it was found that despite the absence of a direct acute toxic effect, TiO_2_ and ZrO_2_ NPs caused significant deviations in biochemical indexes and behavior parameters of rats. Apparently, the observed changes are related to changes in the antioxidant enzyme activity of rats, metabolic disorders, and neurotoxicological and neurodegenerative effects of TiO_2_ and ZrO_2_ NPs. The multidirectional shifts relative to the control of the marker timing of physical and functional maturation probably indicated a chemically induced desynchronization of complex processes of heterogeneous systemogenesis in experimental rats, and the consequences of prenatal exposure to TiO_2_ and ZrO_2_ NPs differed, which allows us to declare different activity of TiO_2_ and ZrO_2_ NPs. Even after the descendants reached the age of relative maturity, modification changes retained intergroup differences. Thus, in Group 1 (TiO_2_ NPs), certain disorders in the sphere of circular rotations, leukopenia and hypocholesterolemia were noted, and in the Group 2 (ZrO_2_ NPs), changes in locomotion in the absence of hematological shifts and signs of impaired hepatic metabolism were detected.

The results obtained will become an important basis for further investigation of the mechanisms of toxic action of TiO_2_, ZrO_2,_ and other industrial nanoparticles and decision-making to ensure safety and reduce their impact on animals and humans. Further research will be related to a detailed study of the mechanisms of the effect of TiO_2_ al ZrO_2_ NPs on the vital signs of rats and the determination of the tolerance level. The collected data will allow us to draw parallels with the human organism and understand how important to introduce regulations on the contact of pregnant women with materials and micromachines based on TiO_2_ and ZrO_2_ NPs.

## Figures and Tables

**Figure 1 micromachines-14-00363-f001:**
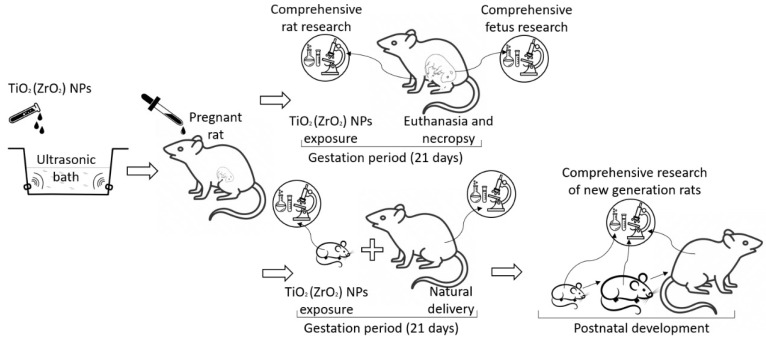
The scheme of the experiment with pregnant rats.

**Figure 2 micromachines-14-00363-f002:**
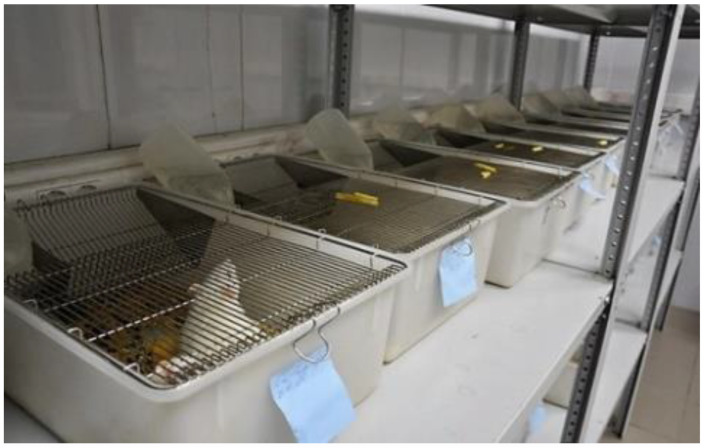
Conditions of keeping laboratory animals.

**Figure 3 micromachines-14-00363-f003:**
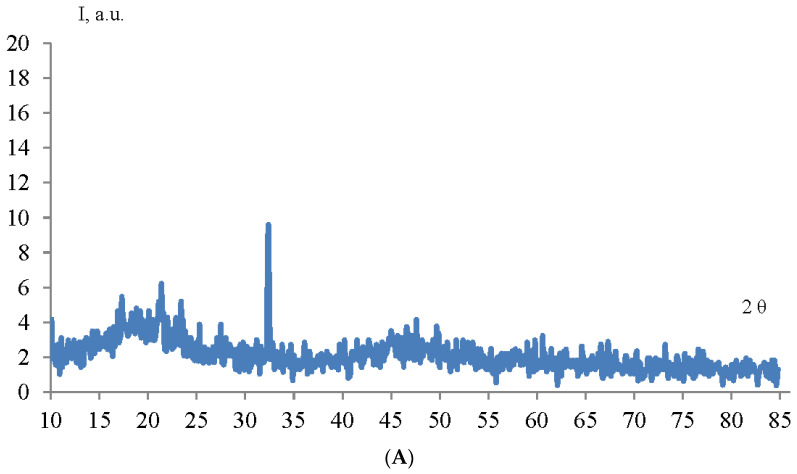
Diffractogram of TiO_2_ (**A**) and ZrO_2_ (**B**) NPs samples.

**Figure 4 micromachines-14-00363-f004:**
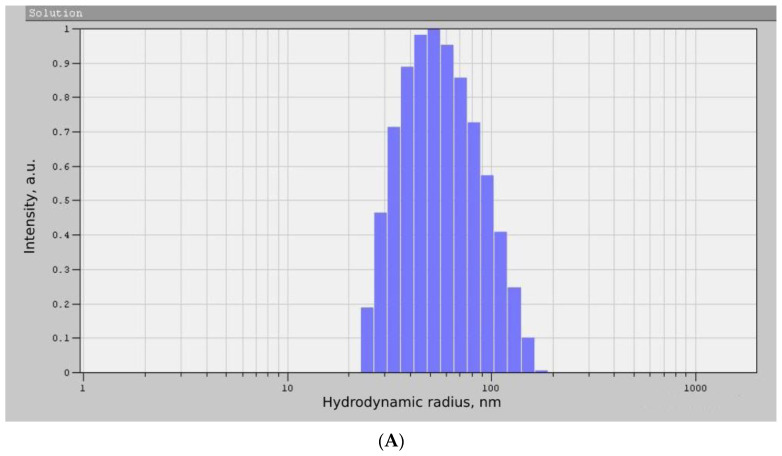
Histogram of the size distribution of TiO_2_ (**A**) and ZrO_2_ (**B**) NPs samples.

**Table 1 micromachines-14-00363-t001:** Results of examination of pregnant female rats exposed to oral exposure to TiO_2_ and ZrO_2_ NPs during the gestational period, *p* < 0.05.

Investigated Indicators	Experimental Groups		
	Group 1 (TiO_2_ NPs)	Group 2 (ZrO_2_ NPs)	Control
Integral physiological indicators			
Body weight gain, g	75.9 ± 6.1	82.1 ± 3.5	83.5 ± 3.0
Summation-threshold indicator, V	2.19 ± 0.09	2.18 ± 0.08	2.16 ± 0.07
Heart rate, bpm	496.7 ± 6.0	487.1 ± 7.7	488.6 ± 7.2
Muscle strength, g	1087.0 ± 42.2	1011.0 ± 48.1	1071.0 ± 56.3
Behavioral reactions over a 5 min observation period			
Duration of movements, s	21.81 ± 4.24	23.74 ± 2.62	16.33 ± 3.25
Time of immobility, s	22.21 ± 6.93	37.20 ± 9.78	64.29 ± 13.92
Duration of hind leg lifts, s	108.72 ± 12.17	109.81 ± 11.56	83.04 ± 12.68
Duration of grooming, s	33.96 ± 9.27	23.34 ± 3.87	16.01 ± 3.16
Frequency of movements	17.4 ± 3.0	21.4 ± 2.4	15.3 ± 2.7
Frequency of acts of immobility	14.4 ± 3.0	16.3 ± 2.8	20.9 ± 3.5
Frequency of hind leg lifts (number of racks)	26.8 ± 3.0	27.6 ± 2.3	22.7 ± 3.3
Frequency of grooming acts	5.0 ± 1.0	5.1 ± 0.8	3.3 ± 0.4
Maximum speed of movement, mm/s	123.67 ± 7.85	140.64 ± 5.14	111.80 ± 8.26
Duration of movement with maximal speed, s	127.33 ± 25.01	94.43 ± 27.06	86.00 ± 24.76
Average speed of movement, mm/s	66.27 ± 1.77	69.29 ± 1.32	66.00 ± 2.07
Distance covered, m	1.394 ± 0.252	1.620 ± 0.163	1.034 ± 0.233
Number of clockwise circular rotations	6.1 ± 0.7	5.4 ± 0.9	5.0 ± 0.9
Number of counterclockwise circular rotations	6.7 ± 1.1	6.9 ± 1.0	4.5 ± 0.9
Average duration of circular rotations, s	8.724 ± 0.433	7.903 ± 0.238	8.737 ± 0.435
Average perimeter of the circle of circular rotations, mm	377.6 ± 31.5	407.7 ± 35.3	334.4 ± 29.1
Average speed of circular rotations, mm/s	4.712 ± 0.427	5.425 ± 0.571	4.115 ± 0.346
Content in peripheral blood			
Leukocytes, 10^9^/L	16.32 ± 1.62	12.72 ± 1.39	11.03 ± 1.28
Erythrocytes, 10^12^/L	5.36 ± 0.20	5.37 ± 0.26	5.59 ± 0.15
Hemoglobin, g/L	101.7 ± 3.5	101.4 ± 5.1	107.6 ± 3.3
Thrombocytes, 10^9^/L	666.9 ± 29.3	621.5 ± 37.3	630.0 ± 35.6
Biochemical indicators			
Aspartate transaminase, µmol/s·L	150.4 ± 13.7	132.6 ± 4.6	149.8 ± 13.6
Alanine transaminase, µmol/s·L	71.10 ± 3.04	73.78 ± 5.13	76.89 ± 4.04
Gamma-glutamyl transferase, units/L	4.379 ± 0.518	5.076 ± 0.668	3.570 ± 0.757
Urea, mmol/L	5.966 ± 0.405	6.390 ± 0.429	6.397 ± 0.413
Pyruvic acid, mmol/L	47.67 ± 1.98	45.13 ± 2.09	43.46 ± 1.26
Glucose, mmol/L	4.169 ± 0.242	3.920 ± 0.252	3.964 ± 0.251
Total protein, g/L	74.87 ± 2.42	72.03 ± 2.03	72.43 ± 0.74
Albumin, g/L	39.86 ± 1.12	40.12 ± 1.45	40.49 ± 0.60
Triglycerides, mmol/L	3.571 ± 0.414	3.882 ± 0.272	4.150 ± 0.360
Cholesterol, mmol/L	2.331 ± 0.269	2.606 ± 0.189	2.926 ± 0.197
Lactic acid, mmol/L	3.949 ± 0.302	3.867 ± 0.210	3.818 ± 0.172
Lactic Acid/Pyruvic acid	82.55 ± 4.54	85.77 ± 2.48	88.48 ± 4.98
Lactate dehydrogenase, units/L	995.3 ± 205.6	683.2 ± 108.7	972.9 ± 114.1
Creatinine, µmol/L	37.50 ± 2.96	33.51 ± 3.95	31.90 ± 2.84
Reduced glutathione, µmol/L	1.274 ± 0.139	1.158 ± 0.134	1.231 ± 0.117
MDA, µmol/L	10.72 ± 0.51	10.41 ± 0.34	12.63 ± 0.52

**Table 2 micromachines-14-00363-t002:** Distribution of pregnant female rats by study stages, *p* < 0.05.

Investigated Indicators	Experimental Groups		
	Group 1 (TiO_2_ NPs)	Group 2 (ZrO_2_ NPs)	Control
Number of evaporated females	25	25	25
Number of pregnant females	22	24	23
Conception rate, %	88.0	96.0	92.0
Number of females left to obtain fetal material	11	10	9
The number of females left to receive live-born offspring	11	14	14

**Table 3 micromachines-14-00363-t003:** Autopsy results of pregnant rats treated with TiO_2_ and ZrO_2_ NPs during the gestational period, *p* < 0.05.

Investigated Indicators	Experimental Groups		
	Group 1 (TiO_2_ NPs)	Group 2 (ZrO_2_ NPs)	Control
Quantity per female			
Yellow bodies	11.9 ± 0.6	11.2 ± 0.4	11.7 ± 0.5
Born fetuses	10.8 ± 0.6	10.6 ± 0.6	10.4 ± 0.7
Intrauterine death, %			
Before treatment	4.2	1.8	3.8
After treatment	5.3	3.6	6.9
Total	9.2	5.4	10.5
Fetal somatometric parameters			
Body weight, g	3.85 ± 0.10	4.00 ± 0.14	3.99 ± 0.10
Craniocaudal size, mm	36.3 ± 0.2	36.3 ± 0.4	36.2 ± 0.2
Characteristics of the fetal life support system			
Placenta weight, g	0.58 ± 0.02	0.64 ± 0.03	0.62 ± 0.03
Diameter of the placenta disc, mm	15.0 ± 0.1	15.2 ± 0.1	15.2 ± 0.1

**Table 4 micromachines-14-00363-t004:** Data of microanatomic analysis of transversal body sections of fetuses from female rats exposed to TiO_2_ and ZrO_2_ NPs during pregnancy, *p* < 0.05.

Investigated Indicators	Experimental Groups		
	Group 1 (TiO_2_ NPs)	Group 2 (ZrO_2_ NPs)	Control
The number and sex of the examined fetuses	29 ♂	26 ♂	27 ♂
	21 ♀	24 ♀	23 ♀
Number of fetuses with embryogenesis disorders	8/50	8/50	4/50
Frequency of individual developmental defects, %			
Hydronephrosis	2.0	2.0	2.0
Nephroptosis	6.0	6.0	2.0
Megaureter	2.0	0	0
Megacystis	2.0	2.0	0
Testicular dystopia	2.0 (3.4% male)	2.0 (3.8% male)	0
Hemoperitoneum	0	2.0	0
Hemorrhages in the liver	2.0	0	0
Hemorrhages in the salivary glands	0	0	2.0
Subcutaneous hematomas	0	0	2.0
Intermuscular hematomas	0	2.0	0
The total frequency of anomalies in the group, %	17.4	17.8	8.0

**Table 5 micromachines-14-00363-t005:** Postnatal mortality and body weight dynamics of offspring of rats exposed to TiO_2_ and ZrO_2_ NPs during pregnancy, *p* < 0.05.

Investigated Indicators	Experimental Groups		
	Group 1 (TiO_2_ NPs)	Group 2 (ZrO_2_ NPs)	Control
Number of newborn rats	97	141	140
Average newborn rate, rats/female	8.8 ± 0.9	10.1 ± 0.5	10.0 ± 0.5
Death in the postnatal period, %			
4th day	9.6	2.1	5.3
7th day	11.5	8.5	5.3
14th day	12.5	12.8	6.9
21st day	12.5	20.6	8.4
1 month	13.5	22.7	9.2
2 months	13.5	22.7	9.2
Body weight gain, g			
14th day	20.8 ± 0.3	18.3 ± 0.3	19.1 ± 0.3
21st day	30.1 ± 0.5	27.6 ± 0.4	28.0 ± 0.5
1 month	61.3 ± 1.1	58.6 ± 1.0	58.6 ± 1.0
2 months	153.7 ± 3.5	150.0 ± 2.9	156.6 ± 2.4

**Table 6 micromachines-14-00363-t006:** Indicators of physical development of offspring of rats exposed to TiO_2_ and ZrO_2_ NPs during pregnancy, *p* < 0.05.

Investigated Indicators	Age, Days	Number of Individuals with the Presence of the Trait		
		Group 1 (TiO_2_ NPs)	Group 2 (ZrO_2_ NPs)	Control
Detachment of the auricle, %	4	81.1	71.7	74.3
	5	98.4	95.4	97.6
	6	100.0	99.2	100.0
Eruption of the lower incisors, %	12	92.3	97.6	97.5
Eye Opening, %	19	100.0	98.8	99.2
	21	100.0	100.0	100.0
Lowering of the testes, %	25	83.7	78.7	57.4
	30	100.0	100.0	100.0
Opening of the vagina, %	45	30.8	39.6	45.3
	55	94.7	78.3	98.4
Muscle strength, g	30	334.5 ± 6.7	321.9 ± 6.7	335.8 ± 6.6
	60	826.3 ± 14.8	808.4 ± 15.0	802.1 ± 12.7

**Table 7 micromachines-14-00363-t007:** Maturation rate in offspring of some sensory-motor reflexes of the lactation period.

Investigated Indicators	Age, Days	Number of Individuals with the Presence of the Trait, %		
		Group 1 (TiO_2_ NPs)	Group 2 (ZrO_2_ NPs)	Control
Flipping on a plane	5	94.7	90.2	87.1
	6	100.0	99.2	93.5
Negative geotaxis	5	38.3	52.3	51.8
	7	78.3	94.6	88.5
Homing	12	53.8	72.8	50.8
	14	83.5	93.5	85.2
Flipping in free fall	19	100.0	98.4	100.0

**Table 8 micromachines-14-00363-t008:** Results of examination of two-month-old offspring (both sexes) of female rats exposed to TiO_2_ and ZrO_2_ NPs during pregnancy (average of 24–40 definitions).

Investigated Indicators	Experimental Group		
	Group 1 (TiO_2_ NPs)	Group 2 (ZrO_2_ NPs)	Control
Integral physiological indicators			
Summation-threshold indicator, V	2.38 ± 0.06	2.36 ± 0.05	2.39 ± 0.05
Heart rate, bpm	505.0 ± 6.4	500.8 ± 6.8	500.5 ± 7.2
Behavioral reactions over a 5 min observation period			
Duration of movements, s	28.25 ± 1.65	25.10 ± 1.63	26.66 ± 1.80
Time of immobility, s	10.51 ± 1.84	20.47 ± 5.56	14.75 ± 3.63
Duration of hind leg lifts, s	161.67 ± 6.44	148.13 ± 8.34	154.71 ± 7.81
Duration of grooming, s	31.89 ± 4.26	25.81 ± 4.38	29.83 ± 4.11
Frequency of movements	25.3 ± 1.4	22.3 ± 1.5	23.1 ± 1.6
Frequency of acts of immobility	6.9 ± 0.9	10.2 ± 1.6	8.5 ± 1.2
Frequency of hind leg lifts (number of racks)	33.6 ± 1.2	30.8 ± 1.6	31.2 ± 1.4
Frequency of grooming acts	5.4 ± 0.5	4.9 ± 0.7	5.3 ± 0.6
Maximum speed of movement, mm/s	162.52 ± 4.43	170.72 ± 4.53	162.53 ± 4.94
Duration of movement with maximal speed, s	100.45 ± 14.62	72.95 ± 12.44	114.15 ± 14.92
Average speed of movement, mm/s	74.68 ± 0.88	76.97 ± 0.91	74.40 ± 0.97
Average speed during the observation period, mm/s	7.96 ± 0.46	7.65 ± 0.48	7.30 ± 0.47
Distance covered, m	2.388 ± 0.139	2.295 ± 0.144	2.191 ± 0.142
Number of clockwise circular rotations	8.6 ± 0.6	8.1 ± 0.8	9.3 ± 0.7
Number of counterclockwise circular rotations	9.6 ± 0.5	8.9 ± 0.6	8.7 ± 0.6
Average duration of circular rotations, s	7.542 ± 0.204	7.249 ± 0.183	7.437 ± 0.169
Average perimeter of the circle of circular rotations, mm	260.7 ± 7.1	249.6 ± 7.0	240.6 ± 6.4
Average speed of circular rotations, mm/s	36.56 ± 1.23	37.71 ± 1.22	34.72 ± 1.01
Content in peripheral blood			
Leukocytes, 10^9^/L	15.23 ± 0.80	16.76 ± 0.78	18.31 ± 1.01
Erythrocytes, 10^12^/L	5.95 ± 0.08	5.90 ± 0.09	5.99 ± 0.10
Hemoglobin, g/L	127.7 ± 2.1	128.2 ± 1.8	128.4 ± 1.9
Thrombocytes, 10^9^/L	542.1 ± 20.5	500.5 ± 21.0	537.4 ± 20.4
Biochemical indicators			
Aspartate transaminase, µmol/s·L	177.7 ± 13.5	190.8 ± 19.2	174.6 ± 13.8
Alanine transaminase, µmol/s·L	121.6 ± 8.6	137.9 ± 9.3	122.7 ± 7.1
Gamma-glutamyl transferase, units/L	5.529 ± 0.710	6.001 ± 0.912	4.828 ± 0.678
Urea, mmol/L	5.792 ± 0.321	6.020 ± 0.348	6.658 ± 0.338
Pyruvic acid, mmol/L	40.86 ± 1.21	37.76 ± 0.91	40.84 ± 1.31
Glucose, mmol/L	8.608 ± 0.157	8.270 ± 0.144	8.371 ± 0.192
Total protein, g/L	68.89 ± 2.38	67.89 ± 2.23	69.90 ± 2.47
Albumin, g/L	42.48 ± 1.43	42.29 ± 1.48	43.15 ± 1.57
Triglycerides, mmol/L	0.594 ± 0.049	0.723 ± 0.052	0.681 ± 0.030
Cholesterol, mmol/L	1.747 ± 0.055	2.013 ± 0.086	1.964 ± 0.078
Lactic acid, mmol/L	3.581 ± 0.179	3.311 ± 0.124	3.530 ± 0.189
Lactic Acid/Pyruvic acid	87.07 ± 2.51	87.50 ± 2.10	86.55 ± 3.10
Lactate dehydrogenase, units/L	681.4 ± 100.0	621.7 ± 63.2	573.2 ± 62.4
Creatinine, µmol/L	34.39 ± 1.16	34.75 ± 1.13	35.08 ± 0.95
Reduced glutathione, µmol/L	1.066 ± 0.029	1.078 ± 0.046	1.156 ± 0.038
MDA, µmol/L	10.82 ± 0.25	10.51 ± 0.22	10.87 ± 0.22

## Data Availability

All data are available upon request from the corresponding author.

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
