# Peer review of "Study of the Embryonic Toxicity of TiO2 and ZrO2 Nanoparticles"

_micromachines, 2023, doi:10.3390/mi14020363_

Round 1

Reviewer 1 Report

The manuscript entitled the Study of the embryonic toxicity of TiO2 and ZrO2 nanoparticles explained the embryonic toxicity of TiO2 and ZrO2 nanoparticles, which is not directly included in the scope of this journal. The authors should add strong points to enhance the capability of this manuscript to adhere to the scope and aim of this journal. In addition, the manuscript is not well organized. Please modify the manuscript considering the following comments. 

·         It is well known about the toxicity of TiO2 and ZrO2 nanoparticles. Please add relevant points that make this manuscript different from the already reported works on this subject.

·         Please highlight the novelty and how this manuscript can fill the existing research gap in the introduction section. Add a paragraph with all these points.

·         Add pictorial representations and cartoons of experimental steps to make the manuscript more attractive to readers.

·         Please add the photographs of the experiment. 

·         Figure 2. Histogram of the size distribution of TiO2 (A) and ZrO2(B) NPs samples. The figure seems low quality, and the upper limit of the peak is invisible. The authors need to re-plot the graph by rescaling at the X and Y axes. Rename the X and Y axis title appropriately.

·         Working solutions of TiO2 and ZrO2 NPs at a dose of 100 mg/kg/day were administered to females using a probe at the rate of 1.0 ml per 100 g of body weight practically throughout the entire gestational period from the 1st to the 20th GD. Does this dose simulate human exposure to TiO2 and ZrO2 nanoparticles in her daily life? Please give statements on selecting this dose to administer to the pregnant female rats in this manuscript.

·         What is the tolerance level? Please add discussions on the maximum and minimum levels of dosses the tested animals could tolerate. Compare your findings with the pre-existing results. Add more discussion. 

·         The conclusion needs to rewrite. The present conclusion just included the listing of results and discussions. Please add the significance of the findings from this manuscript and add points on how it filled the research gap. In addition, include the current problem and future trends.

Author Response

We are grateful to the Reviewer 1 for his/her evaluation and for the time devoted to review our manuscript. All comments were useful and pleased us with the high level of understanding of the topic. We have addressed all recommendations as requested. All changes in the manuscript are marked by green. Please see the point-by-point answers below

The manuscript entitled the Study of the embryonic toxicity of TiO2 and ZrO2 nanoparticles explained the embryonic toxicity of TiO2 and ZrO2 nanoparticles, which is not directly included in the scope of this journal. The authors should add strong points to enhance the capability of this manuscript to adhere to the scope and aim of this journal. In addition, the manuscript is not well organized. Please modify the manuscript considering the following comments. 

  • It is well known about the toxicity of TiO2 and ZrO2 nanoparticles. Please add relevant points that make this manuscript different from the already reported works on this subject.

Thank you for the note. We modified Introduction section and pointed out the relevance and of our work.

  • Please highlight the novelty and how this manuscript can fill the existing research gap in the introduction section. Add a paragraph with all these points.

Thank you for recommendation. We expanded the last paragraph to explain why this study is important and to point out its’ novelty.

  • Add pictorial representations and cartoons of experimental steps to make the manuscript more attractive to readers.

 Thank you for nice recommendation. We agree, that in previous state the design of the experiment was complicated to perceive by readers. We have done our best to show the main plan of the experiment in simple scheme. New figure was added to Materials and Method section. Thank you again for recommendation.

  • Please add the photographs of the experiment. 

Thank you for recommendation. For this experiment we have photographs of prepared rats and photos of rats maintenance. To leave the article attractive to all categories of potential readers we decided to show conditions of rats keeping instead of dissected rats. That is why we initially decided to show all results in tables. New figure was added in Section 2.4.

  • Figure 2. Histogram of the size distribution of TiO2 (A) and ZrO2(B) NPs samples. The figure seems low quality, and the upper limit of the peak is invisible. The authors need to re-plot the graph by rescaling at the X and Y axes. Rename the X and Y axis title appropriately.

Thank you for recommendation. Figure 2 was formed by histograms automatically created by software Photocor Complex (Zelenograd, Russia). Following your kind recommendation, we have done our best to improve the representative quality of histogram. Thus, we added the peak meaning (the maximum intensity is 1). Both X and Y axes were titled. It is important to note that X axis is logarithmic. It is justified by comfortable representation of hydrodynamic radius of particles. That is why we left the origin scale. 

  • Working solutions of TiO2 and ZrO2 NPs at a dose of 100 mg/kg/day were administered to females using a probe at the rate of 1.0 ml per 100 g of body weight practically throughout the entire gestational period from the 1st to the 20th GD. Does this dose simulate human exposure to TiO2 and ZrO2 nanoparticles in her daily life? Please give statements on selecting this dose to administer to the pregnant female rats in this manuscript.

Thank you very much for the nice question. According to EFSA (2016), Weir et al. (2012) and Bachler et al. (2015) adults consume about 1 mg of TiO2/kg bw/day in average. Children can consume by 10 mg of TiO2/kg bw/day and pregnant women can intake 2-3 times more TiO2. However, in this work we did not consider risks of daily consumption of TiO2 or ZrO2 NPs by pregnant women because this is a very variable parameter that depends on many factors. Actually, we reviewed embryo/fetotoxicity and teratogenicity of TiO2 and ZrO2 NPs on laboratory animals. Based on reported by other researcher’s data we set the average dose declared as toxic as for TiO2 as for ZrO2 NPs. In our opinion, the set dose is suitable for the experiment. The most important thing is that the conditions for the experiment for TiO2 and ZrO2 NPs were the same.

To explain how the dose was set we added the corresponding sentence.

Weir A, Westerhoff P, Fabricius L, Hristovski K, von Goetz N (2012) Titanium dioxide nanoparticles in food and personal care products. Environ Sci Technol 46:2242–2250. https://doi.org/10. 1021/es204168d

EFSA ANS Panel (2016). Scientific opinion on the re-evaluation of titanium dioxide (E 171) as a food additive. EFSA J. 14:4545 (83 pp.). https://doi.org/10.2903/j.efsa.2016.4545

Bachler G, von Goetz N, Hungerbuhler K (2015) Using physiologically based pharmacokinetic (PBPK) modeling for dietary risk assessment of titanium dioxide (TiO2) nanoparticles. Nanotoxicology 9:373–380. https://doi.org/10.3109/17435390. 2014.940404

  • What is the tolerance level? Please add discussions on the maximum and minimum levels of dosses the tested animals could tolerate. Compare your findings with the pre-existing results. Add more discussion. 

Thank you for nice recommendation. This manuscript is the first work in the cycle of work on study of toxic effects of TiO2 and ZrO2 NPs. In this work, we have studied many parameters in pregnant rats, embryos, fetuses, offspring, and adult rats of a new generation. To make the experiment reproducible, we reduced the number of variables and provided the same experimental conditions with both TiO2 and ZrO2 NPs. For both NPs exposure concentration was 100 mg/kg/day (the set dose is justified in previous response). Thus, in this work we determined the qualitative and quantitative effects of the same exposure of nanoparticles on the vital signs of laboratory animals in pregnancy, but we could not determine the level of tolerance of animals based on the design of the experiment. However, we agree that tolerance level is very important for determination of any kind of toxicity and will plan to study it in the next big experiment. Thank you again for suitable and useful recommendation.

Also we agree that Discussion part should be improved and we modified it.

  • The conclusion needs to rewrite. The present conclusion just included the listing of results and discussions. Please add the significance of the findings from this manuscript and add points on how it filled the research gap. In addition, include the current problem and future trends.

Thank you for your recommendation. Conclusion was modified.

Reviewer 2 Report

In this study the authors evaluated the effect of TiO2 and ZrO2 NPs embryonic exposure in pregnant rats. They observed that NPs exposure causes few shifts of a similar orientation in the maternal body. Neither TiO2 NPs nor ZrO2 NPs have an embryonic and teratogenic effect on the offspring in utero, but both modify its

postnatal development. 

The study is interesting, and the topic is relevant. Indeed, nanoparticle environmental exposure represents a huge public health issue that will get very relevant in the near future, when people's consciousness on this issue will increase. The authors should take into account the considerations of this reviewer before the paper is accepted. In particular, discussion of novel literature on related topics is required, as well as a better organization of tables to facilitate the readers' understanding.

Minor points:

1) Please have a mention that NPs exposure may lead epigenetic changes in embryons and try to relate this speculation with the data observed (International Journal of Nanomedicine 2016:11 6297–6306).

2) The authors should mention that nanoparticles microenvironmental exposure, including TiO2 exposure, may affect health tissues by inducing changes in biological processes, like autophagy (doi: 10.1016/j.ecoenv.2022.113303), EMT (doi: 10.3390/cancers12010025), metabolism (doi: 10.1016/j.jhazmat.2022.128710, 10.1039/C9EN00137A) which lead chronic inflammation (doi:10.3390/ijerph19010522, 10.1007/s00204-021-02992-7), macrophage recruitment (10.1186/s12989-022-00494-7, 10.1016/j.ecoenv.2021.113112), fibrosis (doi: 10.1080/19338244.2021.2001637, 10.1039/D0EN01021A) and cancer (doi: 10.1016/j.envpol.2022.119293, 10.2147/IJN.S120104). 

3) line 127: correct title: “Determination of of TiO2 and ZrO2 NPs embryotoxicity”

4) In the tables, please highlight that Group 1 corresponds to TiO2 NPs and Group 2 corresponds to ZrO2 NPs. This would help the reader to understand the table easily.

5) The changes reported in embryons after nanoparticle exposure are minor, but sometimes significant. Please, mark with a color when there is an increase (respect to CTRL) or decrease, and maintain the pattern along the various tables. For example, red = increase, green = decrease. This would help the reader to easily associate the changes.

6) Among the many changes reported, only in those reported in some conditions are significant (in many other they are not). Please indicate clearly when the changes (respect CTRL) are significant, maybe indicating the p value.

7) The authors should put more emphasis on possible molecular mechanisms leading to the changes observed and their implications in pathophysiology leading to the phenotypes observed.

8) An evaluation of the NPs toxicity in cell lines derived from embryo (as HUVEC) should be reported.

9) reference 17 is from russian literature, and it will not understandable to the most of scientific community

10) Nanoparticle pollution is a very recent topic which is getting more knowledge in recent years. However, about a third of all references (35) are from literature >5 years. Please limit the oldest references and replace them with more recent studies (when possible).

Author Response

We are grateful to the Reviewer 2 for his/her evaluation and for the time devoted to review our manuscript. All comments were useful and pleased us with the high level of understanding of the topic. We have addressed all recommendations as requested. All changes in the manuscript are marked by green. Please see the point-by-point answers below

In this study the authors evaluated the effect of TiO2 and ZrO2 NPs embryonic exposure in pregnant rats. They observed that NPs exposure causes few shifts of a similar orientation in the maternal body. Neither TiO2 NPs nor ZrO2 NPs have an embryonic and teratogenic effect on the offspring in utero, but both modify its

postnatal development. 

The study is interesting, and the topic is relevant. Indeed, nanoparticle environmental exposure represents a huge public health issue that will get very relevant in the near future, when people's consciousness on this issue will increase. The authors should take into account the considerations of this reviewer before the paper is accepted. In particular, discussion of novel literature on related topics is required, as well as a better organization of tables to facilitate the readers' understanding.

Minor points:

1) Please have a mention that NPs exposure may lead epigenetic changes in embryons and try to relate this speculation with the data observed (International Journal of Nanomedicine 2016:11 6297–6306).

Thank you for recommendation. We added this information in Introduction.

2) The authors should mention that nanoparticles microenvironmental exposure, including TiO2 exposure, may affect health tissues by inducing changes in biological processes, like autophagy (doi: 10.1016/j.ecoenv.2022.113303), EMT (doi: 10.3390/cancers12010025), metabolism (doi: 10.1016/j.jhazmat.2022.128710, 10.1039/C9EN00137A) which lead chronic inflammation (doi:10.3390/ijerph19010522, 10.1007/s00204-021-02992-7), macrophage recruitment (10.1186/s12989-022-00494-7, 10.1016/j.ecoenv.2021.113112), fibrosis (doi: 10.1080/19338244.2021.2001637, 10.1039/D0EN01021A) and cancer (doi: 10.1016/j.envpol.2022.119293, 10.2147/IJN.S120104). 

Thank you very much for recommendation. All suggested works were suitable and supported relevance of the work being mentioned in Introduction.

3) line 127: correct title: “Determination of of TiO2 and ZrO2 NPs embryotoxicity”

Thank you for your attentiveness. We corrected it.

4) In the tables, please highlight that Group 1 corresponds to TiO2 NPs and Group 2 corresponds to ZrO2 NPs. This would help the reader to understand the table easily.

Thank you for recommendation. We mentioned TiO2 and ZrO2 NPs in all tables.

5) The changes reported in embryons after nanoparticle exposure are minor, but sometimes significant. Please, mark with a color when there is an increase (respect to CTRL) or decrease, and maintain the pattern along the various tables. For example, red = increase, green = decrease. This would help the reader to easily associate the changes.

Thank you for the nice recommendation. We introduced color differentiation as was suggested.

6) Among the many changes reported, only in those reported in some conditions are significant (in many other they are not). Please indicate clearly when the changes (respect CTRL) are significant, maybe indicating the p value.

Thank you for recommendation. In Materials and Method section we mentioned that to determine the significance of the difference between groups we used a Student's T-test with the selected significance level α = 0.05 or p<0.05. All differences between groups were significant under set conditions. To avoid adding a new column with p-value in each table we decided to state “p<0.05” in title of tables.

7) The authors should put more emphasis on possible molecular mechanisms leading to the changes observed and their implications in pathophysiology leading to the phenotypes observed.

Thank you for recommendation. We expanded discussion of data obtained and added more relevant references supporting our conclusion.

8) An evaluation of the NPs toxicity in cell lines derived from embryo (as HUVEC) should be reported.

Thank you very much for nice recommendation. Current experimental design of this work didn’t include such study, but we plan to carry out it in the next works. There are still many indexes and parameters that should be studied and compared to completely understand the mechanisms of TiO2 and ZrO2 NPs effect on pregnant rats, embryo, fetus, adult rats.  Nevertheless, we thank you for this advice, as it indicates a good understanding of the topic.

9) reference 17 is from russian literature, and it will not understandable to the most of scientific community

Sorry for this occasion. We added another source.

10) Nanoparticle pollution is a very recent topic which is getting more knowledge in recent years. However, about a third of all references (35) are from literature >5 years. Please limit the oldest references and replace them with more recent studies (when possible).

Thank you for your note. For this topic, there is no many related work, especially regarding TiO2 and ZrO2 NPs and their effect on pregnant animals. That is why we used all sources from high-impact journals that we found. Now with your kind suggestions we expanded Introduction and reached References list with novel works.  

Round 2

Reviewer 1 Report

Accept the manuscript for publication after minor revision.

The manuscript entitled "Study of the embryonic toxicity of TiO2 and ZrO2 nanoparticles" has great importance, and the authors made relevant corrections in the modified manuscript. However, the conclusion seems lengthy. I recommend reducing the size of the conclusion part and making it more concise.

Author Response

Thank you for the positive evaluation of the revised manuscripts. 

We modified Conclusions as was recommended.